# Feasibility randomised controlled trial of a guided workbook intervention to support work-related goals among cancer survivors in the UK

Elizabeth A Grunfeld,[1] Lauren Schumacher,[2] Maria Armaou,[2] Pernille L Woods,[1] Pauline Rolf,[2] Andrew John Sutton,[3] Anjali Zarkar,[4] Steven S Sadhra[5]

For numbered affiliations see end of article.

**Correspondence to**
Professor Elizabeth A Grunfeld; e.grunfeld@bbk.ac.uk

## ABSTRACT

**Objectives** Employment following illness is associated with better physical and psychological functioning. This study aimed to assess the feasibility and acceptability of a theoretically led workbook intervention designed to support patients with cancer returning to work.

**Design** Parallel-group randomised controlled trial with embedded qualitative interviews.

**Setting** Oncology clinics within four English National Health Service Trusts.

**Participants** Patients who had received a diagnosis of breast, gynaecological, prostate or colorectal cancer and who had been receiving treatment for a minimum of two weeks.

**Intervention** A self-guided WorkPlan workbook designed to support patients with cancer to return to work with fortnightly telephone support calls to discuss progress. The control group received treatment as usual and was offered the workbook at the end of their 12-month follow-up.

**Outcome measures** We assessed aspects of feasibility including eligibility, recruitment, data collection, attrition, feasibility of the methodology, acceptability of the intervention and potential to calculate cost-effectiveness.

**Results** The recruitment rate of eligible patients was 44%; 68 participants consented and 58 (85%) completed baseline measures. Randomisation procedures were acceptable, data collection methods (including cost-effectiveness data) were feasible and the intervention was acceptable to participants. Retention rates at 6-month and 12-month follow-up were 72% and 69%, respectively. At 6-month follow-up, 30% of the usual care group had returned to full-time or part-time work (including phased return to work) compared with 43% of the intervention group. At 12 months, the percentages were 47% (usual care) and 68% (intervention).

**Conclusions** The findings confirm the feasibility of a definitive trial, although further consideration needs to be given to increasing the participation rates among men and black and ethnic minority patients diagnosed with cancer.

**Trial registration number** ISRCTN56342476; Pre-results.

## INTRODUCTION

Almost half of adult cancer survivors are of working age,[1] yet patients with cancer are 1.4 times more likely to be unemployed than

### Strengths and limitations of this study

► The study assessed the feasibility and acceptability of a randomised controlled trial (RCT) of a theory-led intervention to support return to work among patients with a diagnosis of breast, gynaecological, colorectal or prostate cancer.

► The intervention used a workbook format comprising paper-based exercises and the development of a return to work plan.

► A mixed method design, with nested qualitative interviews, was used to assess the acceptability of the intervention and the feasibility of the RCT and to determine the utility of the patient-reported outcome measures.

► Only four cancer types were included that may limit generalisability.

► Views of male participants, as well as black and ethnic minority participants, were under-represented, as the majority of participants were female and Caucasian, and all were English speaking.

healthy individuals.[2] Patients with cancer may experience ongoing negative outcomes from the disease or treatment (including pain, fatigue and low mood) that can impact everyday functioning, including work.[3 4] Return to work rates vary across cancer types[5]; however, longer return to work times are associated with certain treatments (eg, chemotherapy[6]), fatigue[7] or a non-supportive work environment.[8] Predictors of return to work include optimal symptom management (as over a quarter of cancer survivors report high symptom burden 1 year postdiagnosis, even after the end of treatment[9]), implementing appropriate workplace adaptations, as well as specific cancer (ie, beliefs about the consequences of cancer) and treatment-related perceptions (ie, beliefs about controlling the effects of cancer at work).

Employment is important not only for individual financial and societal economic

reasons[10] but because being out of work is thought to contribute to, and aggravate, adverse health outcomes.[1 11] Returning to and staying in work following illness is associated with better physical and psychological functioning. Not working is associated with reduced self-esteem, lowered self-efficacy and decreased belief in one's ability to return to the workplace.[12] Furthermore, work is an important component of quality of life,[13] and impaired work is associated with increased depression and anxiety among patients with cancer.[14]

Several interventions to support working have been developed across illness groups, including musculoskeletal disorders, back pain and multiple sclerosis. These interventions have tended to focus on ergonomic adaptation within the workplace with the aim of minimising the risk of physical injuries, likely to be experienced by these patient groups. Interventions targeted at patients with cancer include a 12-week occupational physician-led intervention focused on increasing physical activity to support return to work[15]; a case management approach involving signposting/referring patients to services (eg, physiotherapy, occupational or psychological therapy) to support return to work[16]; and a tool that cancer survivors use to guide discussions about working.[17] However, this tool focused on interactions with employers and healthcare professionals and not on patients' beliefs and barriers that impact workability (one's perception of one's ability to work) and influence work behaviour. Furthermore, a Cochrane review reported low quality evidence for return to work rates for psychoeducational interventions (interventions that encompass a broad range of activities that combine educational and other activities such as counselling and supportive care); however, this was based on only two studies: one that focused on teaching self-care behaviours to manage fatigue and one comprising lectures focused on side effects, stress and coping. The review concluded that there was a need for more high-quality randomised controlled trials (RCTs).[18] Furthermore, a meta-synthesis of qualitative research studies highlighted the need for vocational interventions with patients with cancer to be person-centred and for such interventions to acknowledge the role of social, clinical and work-related factors.[19]

The WorkPlan intervention is theoretically led and uses the self-regulation model[20] and goal-setting theory.[21] WorkPlan was developed following the Medical Research Council (MRC) guidelines for the development of complex interventions[22] and used an intervention mapping methodology for designing and implementing complex interventions or programmes. WorkPlan differs from other published interventions in that it supports people diagnosed with cancer to prepare for returning to work by creating a space to envisage and construct a future at work, then supporting patients to develop appropriate communication and planning skills to support returning to work. The workbook comprises activities aimed at eliciting beliefs about the impact of cancer and of the person's perceived workability, identifying actions to facilitate the process of returning to work and to support specific tasks once within the workplace. The individual then outlines concrete steps to achieve their goals through a structured return to work plan.

## AIMS
The primary objective of the study was to trial the Work-Plan intervention and data collection materials to determine if the materials were acceptable to participants and whether participants were able to provide full answers. The aims were to:
1. Identify whether the materials and intervention were acceptable and understandable.
2. Determine whether the recruitment target was achievable and identify the most successful methods of recruitment.
3. Determine the acceptability of the randomisation process.
4. Identify retention rates in both arms.
5. Determine if data were obtainable to enable a full cost-effectiveness analysis in a definitive trial.

## METHODS
A full description of the protocol is available elsewhere.[23]

### Participants
The target was to recruit 60 participants (30 randomised into each group). This was not a hypothesis testing trial, and the sample size was based on pragmatic assumptions around feasible recruitment figures and the number of participants required to estimate the key parameters around the feasibility of a full RCT.

#### Inclusion criteria
Inclusion criteria include: (A) received a diagnosis breast, gynaecological, prostate or colorectal cancer; (B) had not been classified as having metastatic disease or recurrence; (C) at least 2 weeks post-treatment initiation; (D) aged 18–70 years; (E) working at the time of diagnosis; and (F) and not working at time of recruitment but intended to return to work.

### Recruitment and randomisation
Participants were recruited by researchers and research nurses through four English National Health Service Trusts UK. Participants were identified through cancer clinics and multidisciplinary team meetings and by placing posters and leaflets in clinics, support and information services, chemotherapy suites and CT scan waiting areas. All recruitment sites were based in tertiary care centres. In addition, collaborating clinicians were provided with leaflets and information packs outlining the study. We translated recruitment materials into the five most commonly spoken languages among people of working age in Birmingham (2011 Census): Bengali, Chinese (standard), Polish, Punjabi and Urdu. Funding was available to provide interpreters if required, and

we were able to translate the workbook into different languages if required by participants. Potential participants who expressed interest in the study were provided with a study information sheet and asked to provide their contact details for the researcher to phone and confirm if they were interested in participating. Eligible participants who expressed an interest in participation were screened over the phone and were sent an invitation for an assessment interview (which included an explanation of the randomisation process) at the hospital or over the telephone. All participants were required to provide written consent to participate.

Following the assessment interview the researchers used an online and text-based randomisation system (Sealed Envelope Ltd) to randomise participants at a ratio of 1:1 between the intervention group and usual care group. Participants were stratified by age (18–50 years or 51 years and over) and cancer type (breast, bowel, gynaecological or urological). Participants who consented into the study were logged according to recruitment site using the National Institute for Health Research's Central Portfolio Management System.

### Intervention
The WorkPlan package is a 4-week guided workbook intervention consisting of structured sections and activities to provide guidance and support to patients. The workbook is broken down into chapters. Chapter one focused on thinking about illness and treatment (based around the illness perceptions component of the Self-Regulation Model) and includes causes of cancer, symptoms, beliefs about efficacy of treatment and consequences of living as a survivor of cancer. It then explores the participant's beliefs about the impact of cancer and treatment on their ability to function in the workplace, including suggestions for management. The chapter concludes by examining participants' emotional reactions to treatment and support/strategies to manage these. Chapter two is focused on setting and achieving goals (based on Goal Theory) including the goal-setting process, identifying and overcoming barriers and using support. Chapter three works on building confidence, including ways to boost confidence. This chapter concludes by examining fatigue and ways to to identify and manage fatigue triggers. Chapter four focuses on developing an action plan for returning to work and outlines how to initiate discussions and deal with difficult questions within the workplace.

Participants were encouraged to work through chapters in turn during each week of the intervention period, allocating around 120 min per week. However, this was not strictly monitored, and participants had the opportunity to work through the intervention at a pace that suited them and their timeframe for returning to work. Participants incorporated all elements from the workbook into a personal *return to work plan*, which they were encouraged to develop in the final week. A resources section signposted participants towards relevant avenues

of further support. Multiple copies of the return to work planning page were available to encourage changes to be made when necessary, and these plans were used as a tool when meeting with employers to aid discussion around returning to work. Service users in the original pilot work (of the materials and study design) were concerned about raising work-related issues too early with their employer and stated that they would prefer to engage with their workplace after completing the intervention, when they felt better able to represent their view and had formed a return to work plan. Therefore, the intervention does not have a specific employer component, but rather the workbook promotes skills to enable communication with employers.

### Control group
Participants received usual care, which focused on clinical care and optimal symptom management. In order to prevent participants from undertaking activities in the workbook, the information sheets and prerandomisation discussion did not include the content or focus of the intervention, and participants were not offered the workbook until after their 12-month follow-up.

Participants in both groups were able to access other information and support relating to work and were therefore asked to record any resources or information they used during the trial.

### Study outcomes
#### Primary outcome
The primary outcomes were time to return to work and return to work rates at 6-month and 12-month follow-ups. Any changes in working status, for example, contracted hours and duties, were documented along with specific reasons for non-return to work (eg, unavailability of job and ongoing medical concerns) to determine whether to incorporate specific reasons for non-return as measures in a full trial.

#### Secondary outcomes
Secondary outcome measures included mood, satisfaction with return to work and satisfaction with the return to work process.

Data were collected at four time points: baseline (T0), 4-week postintervention (intervention group) or 4-week postrandomisation (usual care group) (T1) and 6-month postrandomisation (T2) and 12-month postrandomisation (T3) follow-ups. At each time point, participants were mailed a questionnaire pack (with a prepaid self-addressed envelope) that comprised: (1) Illness Perceptions Questionnaire-Revised[24]; (2) Brief Illness Perception at Work Scale[25]; (3) Hospital Anxiety and Depression Scale[26]; (4) Work Ability Index[27]; (5) satisfaction with return to work if returned to work (single item); (6) Satisfaction with Work Scale[28] (if returned to work); (7) EuroQoL-5D-5L (EQ-5D-5L) (quality of life)[29]; (8) visual analogue scale measure of quality of life (single item).[30]

## Work status and healthcare utilisation

Participants provided details of their use of services and employment activity via text message (using JANET; http://www.pageone.co.uk/services/janet-txt). A maximum of four text messages were sent to participants at the end of each month to gather information on their work status (ie, full-time, part-time, phased return, sick leave and not working), number of days worked that month and healthcare utilisation (number of general practitioner appointments that month). Monthly intervals were chosen as memory of general practitioner appointments is around 4 weeks, so we could not rely on accurate recall of healthcare utilisation at 6-month questionnaire follow-ups.[31 32]

## Adverse events

A record sheet was produced for the recording of adverse events and provided as part of the study file to all participating sites. If participants reported any negative events, including low mood or anxiety, as a result of taking part in the study or undertaking the intervention, or if participants withdrew from the study due to an adverse event, then these were recorded on the record sheet.

## Qualitative interviews

The aim of the postintervention and 12-month follow-up interviews was to better understand the effects of the intervention and to explore how the intervention was experienced by participants. Participants were approached sequentially until the recruitment target was reached. Interviews were conducted over the telephone or face to face, depending on the participant's preference. The postrandomisation interview focused on: (1) beliefs about work and cancer; (2) experience of employment and work values; (3) ways in which returning to work could be supported; and (4) expectations of the Work-Plan intervention. Twelve-month interviews explored: (1) beliefs about cancer and work and how these were challenged over the preceding year, (2) general perceptions of the trial and (3) the personal return to work process of each individual.

## Data analysis

Descriptive statistics for all between group outcome measures are presented, including means (SD) and frequencies. The purpose of this feasibility study was not hypothesis testing, and the sample size was underpowered to undertake the full analysis that would be used in a full trial (analysis of covariance adjusting for baseline values).

### Economic analysis

Although an economic evaluation was not suitable in the context of a feasibility trial, we did aim to determine whether data would be obtainable to enable a full cost-effectiveness analysis in a definitive trial.

### Qualitative data

Interviews were audio-recorded, transcribed verbatim and analysed using the framework method[33] to identify emergent themes. The Consolidated Criteria for Reporting Qualitative Research was used to guide the presentation of the qualitative analyses. Findings from the quantitative and qualitative analyses are presented concurrently, although fuller details of the methodology for the interviews and of the qualitative findings have been reported elsewhere.[34]

## Patient and public involvement

The WorkPlan intervention was developed following a prospective study, which followed patients with cancer for 1 year following the end of treatment to identify factors that influenced the likelihood of not returning to work. The WorkPlan intervention was then refined with input from 15 cancer survivors who provided feedback on the content and wording of the workbook and format of delivery (workbook or face-to-face sessions). One steering group member and author (PR) was a patient representative and provided input on recruitment, study design and materials. Study participants were asked if they would like a lay summary of the study findings at the end of the study, and this was sent to all who indicated that they would.

## RESULTS

### Eligibility and recruitment

During the recruitment period, 324 patients were identified and assessed for eligibility and 170 participants were ineligible (reasons included not working at time of diagnosis, older than 70 years or already returned to work). A further 154 patients were considered eligible (figure 1). Eighty-six of these participants declined to participate resulting in 68 participants (44%) being consented and randomised into the study. Although 68 participants consented to take part in the study, only 58 returned

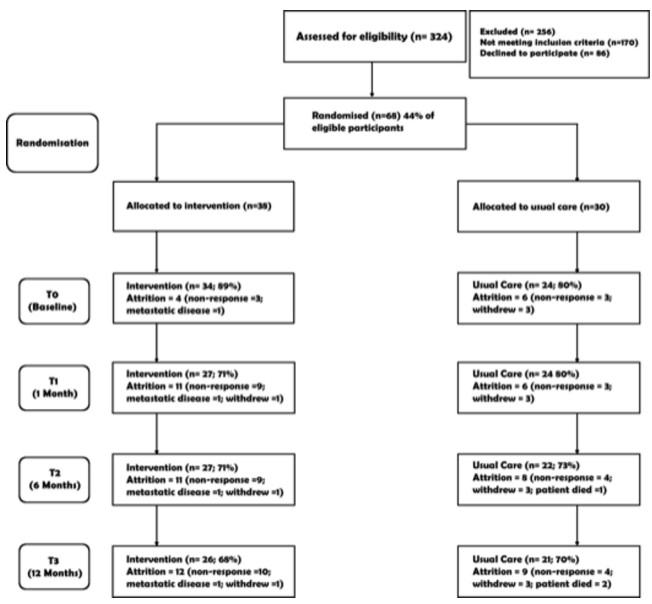

**Figure 1** Flow of participants through the study (showing cumulative attrition at each time point from point of randomisation).

fully completed T0 (baseline) questionnaires. There was no difference between those who returned the questionnaires and those who did not on any of the demographic or clinical measures, site of recruitment, size of employer or randomised group. In the interviews participants reported that the randomisation process was acceptable.

Furthermore, 23 participants in the intervention group were interviewed at the postintervention (T1) time point, which exceeded our target of 20. At the 12-month (T3) follow-up, 22 participants from the intervention group and 20 from the control group participated in an interview, which exceeded our target of 40 participants in total.

## Sample characteristics

We were unable to collect demographic information of participants who chose not to consent into the study. The majority of participants self-identified as being white (88%), followed by African/Afro-Caribbean (7%) and Asian (5%) (table 1). There was significantly more women (79%) than men (21%) recruited into the study. This reflected the large percentage of the participants who had been diagnosed with breast cancer (50%) and gynaecological cancers (15%) compared with prostate (16%) and colorectal (19%) cancers. The majority of

| Table 1 Participant demographics | | | |
|---|---|---|---|
| | Usual care (n=30) | Intervention (n=38) | All (n=68) |
| Age, mean (range) | 51.2 (35–63) | 50.4 (25–65) | 50.8 (25–65) |
| Gender, n (%) | | | |
| Female | 26 (87) | 28 (74) | 54 (79) |
| Male | 4 (13) | 10 (26) | 14 (21) |
| Marital status, n (%) | | | |
| Married or living with partner | 20 (67) | 28 (74) | 48 (71) |
| Divorced or separated | 6 (20) | 6 (16) | 12 (18) |
| Single and never married | 4 (13) | 4 (10) | 8 (11) |
| Dependent children living at home, n (%) | 13 (43) | 13 (34) | 26 (38) |
| Ethnicity, n (%) | | | |
| White | 25 (83) | 35 (90) | 60 (88) |
| Asian | 2 (8.5) | 1 (3) | 3 (5) |
| African/Afro-Caribbean | | | |
| Highest education level, n (%) | 2 (8.5) | 3 (7) | 5 (7) |
| Did not complete secondary education | 0 (0) | 1 (3) | 1 (1) |
| Secondary education (to 16 years) | 14 (47) | 9 (24) | 23 (34) |
| Further education (to 18 years) | 3 (10) | 7 (18) | 10 (15) |
| Higher education (degree or higher) | 13 (43) | 21 (55) | 34 (50) |
| Cancer diagnosis, n (%) | | | |
| Breast | 16 (53) | 18 (46) | 34 (50) |
| Urological | 2 (7) | 9 (24) | 11 (16) |
| Bowel | 7 (23) | 6 (16) | 13 (19) |
| Gynaecological | 5 (17) | 5 (14) | 10 (15) |
| Comorbidities, n (%) | | | |
| 1 | 8 (27) | 8 (21) | 16 (24) |
| 2 | 1 (3) | 0 (0) | 1 (1) |
| 3 or more | 6 (20) | 5 (13) | 11 (16) |
| Flexible working allowed, n (%) | 7 (23) | 18 (48) | 25 (37) |
| Number of months entitled to full sick pay, mean (SD) | 3.9 (2.7) | 4.8 (3.2) | 4.4 (2.9) |
| Work status at 6-month follow-up, n (%) | | | |
| Working full/part time* | 9 (30) | 16 (43) | 25 (37) |
| Work status at 12 -month follow-up, n (%) | | | |
| Working full/part time* | 14 (47) | 26 (68) | 40 (59) |

*Including phased return to work.

participants were married or living with a partner (71%), and half were educated to degree level of above (50%). Comorbidities (defined as any concurrent diagnosis of a physical or psychological disorder for which participants were currently receiving treatment) were more common among the usual care (50% reporting one or more comorbidities) than the intervention group (29%), and this could be explored in a future trial as a possible mediator of returning to work.

## Data collection
### Attrition
Attrition rates (determined by non-return of the questionnaire at that time point) from point of randomisation to 1 month (T1), 6 months (T2) and 12 months (T3) follow-up were 24.5% (29% in intervention and 20% in control group), 28% (29% in intervention and 27% in control group) and 31%, respectively (32% in intervention and 30% in control group). This was slightly higher than the conservative 25% estimate we had forecast at 12 months. Attrition was higher when participants were recruited into the trial by research nurses (37% at T3) rather than the study research assistants (19% at T3), potentially indicating that participants are more likely to remain in a study when they had developed a relationship (through the information and consent process) with a researcher on the project.

### Acceptability of the outcome measures and data collection methods
For the most part, the outcome measures were acceptable, and no participants raised study burden as a concern during the interviews. Participants who returned the questionnaires at each time point mostly returned fully completed questionnaires and did not systematically leave questions unanswered. We trialled using a text-based data collection system for collecting monthly work data and general practitioner (GP) visits. This was acceptable to the majority of participants; only two participants did not have access to a mobile phone and so we used monthly emails to collect these data with these participants.

### Adverse events
No adverse events were reported during the feasibility trial.

### Participants views of the intervention
From the interviews, it was apparent that overall participants enjoyed taking part in the intervention and that it provided a focus and clarity regarding the process of returning to work and options that they might consider. The workbook was described as a useful tool that facilitated the planning process for returning to work and that the exercises within the workbook 'broke it [the process] down into small bits'. It allowed participants to imagine the potential problems around roles, tasks and events that could arise and plan how to deal with these. Furthermore, by considering interactions in the workplace, such as coping with coworkers' reactions, they were able to

engage in mental role play to rehearse how to respond, gaining confidence in managing a successful return to work.

The workbook format of the intervention was well received by participants and preferred by the majority of participants to the idea of using an online or app version. The booklet was seen as convenient (easy to access compared with an online version), simple to transport and could easily be shared with others.

> I think the benefit of doing the booklet with a pen in your hand is that you actually feel more engaged with it. You're writing it and you're physically owning your words. Whereas on the computer I think it would be a lot more impersonal. (P24, aged 54 years)

> I've been able to go to bed and I've been able to read it. With my phone it's small, do you know what I mean? Things could be missed. If it's a book where you come back to it now and again. It's there at the side of my bed and kind of prompts you to do it. (P10, aged 44 years)

However, a minority of interviewees did suggest that it would be more secure (requiring a password) and environmentally friendly to have an online version.

Although the intervention was perceived as useful, some interviewees did highlight areas that could either have been more detailed in the booklet or that had not been included. The main area where participants required greater detail was about managing finances during sick leave periods and about financial support that could be available to them. This is an area of the current intervention that would need to be adapted before moving to a full trial. Furthermore, during the interviews, one participant commented on the wording of the booklet, which referred to patients having completed treatment, whereas the patient was still going through treatment, and this made it difficult for them to engage suggesting that the booklet might need rewording.

> I was still trying to go through treatment and it was talking about the process when it had all finished. I was still going through treatment, so I remember being quite annoyed about that. (P15)

### Outcomes measures
The primary outcomes were time to return to work and return to work rates at 6-month and 12-month follow-ups. There were no significant differences between the usual care and intervention group in terms of the number of days from leaving work to returning to work (usual care mean 308 days (SD 74) compared with intervention mean of 333 days (SD 153)). The greater number of days observed for the intervention group was likely influenced by the fact that within the intervention group (mean 190 days, SD 145), there were a greater number of days (non-significant) between leaving and consenting into the study (so this group had already been out of work for a longer period of time) compared with the usual care

**Table 2** Outcome measures by assessment time point and group

| | Study group | T0 baseline, mean (SD) | T1 postintervention, mean (SD) | T2 6-month follow-up, mean (SD) | T3 12-month follow-up, mean (SD) |
|---|---|---|---|---|---|
| Emotion (IPQ-R) | Usual care | 2.7 (0.6) | 2.6 (0.7) | 2.4 (0.5) | 2.6 (0.6) |
| | Intervention | 2.9 (0.6) | 2.6 (0.7) | 2.6 (0.5) | 2.7 (0.7) |
| Timeline (IPQ-R) | Usual care | 3.3 (0.1) | 3.3 (0.6) | 3.1 (0.5) | 3.0 (0.5) |
| | Intervention | 2.9 (0.1) | 3.1 (0.7) | 3.1 (0.6) | 3.0 (0.8) |
| Illness coherence (IPQ-R) | Usual care | 3.3 (0.5) | 3.3 (0.6) | 3.4 (0.6) | 3.5 (0.6) |
| | Intervention | 3.1 (0.7) | 3.1 (0.7) | 3.1 (0.7) | 3.2 (0.8) |
| Illness consequences (IPQ-R) | Usual care | 3.9 (0.7) | 3.9 (0.5) | 3.9 (0.3) | 3.9 (0.3) |
| | Intervention | 3.9 (0.7) | 4.0 (0.4) | 3.8 (0.4) | 3.9 (0.4) |
| Personal control (IPQ-R) | Usual care | 3.4 (0.8) | 3.4 (0.4) | 3.4 (0.4) | 3.2 (0.4) |
| | Intervention | 3.3 (0.8) | 3.3 (0.6) | 3.3 (0.4) | 3.2 (0.7) |
| Treatment control (IPQ-R) | Usual care | 3.5 (0.4) | 3.5 (0.4) | 3.5 (0.4) | 3.4 (0.4) |
| | Intervention | 3.5 (0.4) | 3.4 (0.4) | 3.5 (0.3) | 3.5 (0.4) |
| Brief illness perception at work scale | Usual care | 3.6 (0.5) | 3.7 (0.9) | 3.6 (0.9) | 3.4 (0.5) |
| | Intervention | 4.0 (0.9) | 3.6 (0.7) | 3.4 (1.1) | 3.6 (0.9) |
| EQ-5D-5L (health status) | Usual care | 2.1 (0.1) | 1.8 (0.6) | 1.7 (0.5) | 1.6 (0.5) |
| | Intervention | 1.7 (0.8) | 1.6 (0.5) | 1.6 (0.4) | 1.5 (0.4) |
| Visual analogue scale measure of quality of life | Usual care | 56.2 (18.7) | 59.3 (20.7) | 71.6 (17.9) | 75.0 (19.5) |
| | Intervention | 61.6 (19.3) | 68.0 (22.6) | 73.7 (14.7) | 77.8 (17.1) |
| Anxiety (HADS) | Usual care | 8.3 (4.3) | 7.9 (4.4) | 7.2 (4.5) | 6.7 (3.4) |
| | Intervention | 7.2 (4.3) | 7.3 (4.5) | 6.3 (3.5) | 6.1 (4.2) |
| Depression (HADS) | Usual care | 5.8 (3.0) | 5.0 (3.8) | 4.9 (3.6) | 4.5 (4.3) |
| | Intervention | 6.1 (4.1) | 5.6 (4.2) | 4.7 (3.5) | 3.9 (3.1) |
| Work ability index (overall) | Usual care | 4.3 (0.7) | 4.4 (3.1) | 6.7 (2.8) | 7.6 (2.6) |
| | Intervention | 2.7 (0.5) | 5.4 (3.7) | 6.6 (3.4) | 7.8 (3.0) |
| Work ability index (physical demands of role) | Usual care | 2.5 (0.3) | 2.8 (1.1) | 3.2 (1.0) | 3.8 (1.0) |
| | Intervention | 2.3 (0.5) | 3.2 (1.2) | 3.3 (1.3) | 3.9 (1.2) |
| Work ability index (cognitive demands of role) | Usual care | 2.7 (0.2) | 2.9 (1.2) | 3.1 (1.3) | 3.7 (1.2) |
| | Intervention | 2.5 (0.2) | 3.0 (1.3) | 3.2 (1.2) | 3.9 (1.2) |
| Satisfaction with work scale | Usual care | 3.7 (0.7) | 3.5 (0.6) | 3.5 (0.7) | 3.4 (0.6) |
| | Intervention | 3.4 (0.9) | 3.3 (1.0) | 3.3 (1.1) | 3.3 (1.0) |
| Number of days worked that month | Usual care | – | 3.5 (7.7) | 5.7 (8.7) | 11.5 (10.1) |
| | Intervention | – | 3.4 (6.5) | 8.9 (9.2) | 12.4 (7.9) |

EQ-5D-5L, EuroQoL-5D-5L; HADS, Hospital Anxiety and Depression Scale; IPQ-R, Illness Perceptions Questionnaire - Revised.

group (mean 158 days, SD 100). Although the intervention group reported a greater number of days worked per month at 6-month and 12-month follow-up (table 2), these did not reach significance, possibly due to lack of power. In addition, at 6-month follow-up, 30% of participants in the usual care group had returned to full-time or part-time work compared with 43% of the intervention group. At 12 months, the percentages were 47% (usual care) and 68% (intervention).

Regarding secondary outcome measures, the intervention group reported less anxiety and depression-related symptoms, although these did not reach significance and

would need to be examined as part of a fully powered RCT. There was potentially a floor effect with the EQ-5D (health status) with few symptoms reported at each timepoint, resulting in low scores across participants with limited dispersion of the scores.

## DISCUSSION

This study aimed to assess the feasibility and acceptability of an RCT of a workbook intervention to support patients with cancer in returning to work. Our results indicate that the format of the workbook was well received and that

the exercises within the workbook were engaging and useful in the return to work process. However, it should be noted that this may be a consequence of acquiescence bias or lack of awareness of the utility of an alternative to format used within the trial. An unexpected outcome was the degree to which participants valued the process of writing within a physical workbook. The workbook aided them in organising their thoughts, enabling them to plan for the future. Expressive writing, or the formation of a written narrative exploring the emotional aspects of a personal experience, allows an individual to organise own thoughts and emotions into a coherent narrative, or summary, that can facilitate more effective coping.[35 36] Using the WorkPlan workbook was not intended to act as an expressive writing task; however, it enabled participants to explore thoughts and emotions related to their cancer and to work and the work environment including organisational (support and shift patterns) aspects of their work, a process that participants identified as both supportive and enjoyable. There is a clear need for an intervention, such as WorkPlan, to support patients in planning their return to work and to support them in thinking about how they could overcome potential issues associated with work tasks, their role (responsibilities) as well accommodating ongoing medical care and treatment (ie, fitting work requirements around medical consultations and treatment).

The recruitment rate was 44%, which was acceptable and in-line with other studies. However, there were fewer men recruited (21%) than we had expected, which was due in part to the cancer types that we recruited into the study with two-thirds of participants having been diagnosed with breast or gynaecological cancers. In a future trial, we would widen the range of cancer types that would be eligible for inclusion and stratify by gender as well as age. The majority of participants also identified as being white (88%); however, within the Birmingham area (where the majority of recruitment took place), around 81% of the population were classified as white,[37] and therefore the findings may not be representative of patients with cancer returning to work after treatment. A future trial will need to make male and minority recruitment a priority and recruitment materials may need amending to be more relevant to these groups.[38] In addition, although we translated recruitment materials into five languages to support recruitment of non-English speakers into the study, we were unsuccessful in recruiting any non-English speakers. However, the main barrier to recruiting non-English speakers was not necessarily a language barrier but rather that that the majority of those approached were not working at the time of diagnosis. Finally, half of study participants were educated to degree level and, although this is in-line with numbers of school leavers entering tertiary education in the UK today, it is higher than would have been observed among the age groups included in this study, and therefore further work is needed to understand the reason for low uptake into the study of people leaving education at an earlier age.

Overall, the study design, using a nested qualitative evaluation, was found to be feasible. A key strength of this study was the mixed methods approach, which allowed for triangulation of experiences of the participants. Qualitative methods provided depth of understanding and the findings, particularly exploration of trial acceptability, suggested that participants were generally satisfied with the process and found the experience acceptable and informative. However, patients may have been inclined to produce socially desirable responses, although with attrition rates within tolerable limits, there is additional support for the acceptability of the study process. Finally, the insights emerging from this study relied on in-depth qualitative investigation; however, it should be noted that the findings are limited to predominately female Caucasian participants.

This was a feasibility study and so efficacy testing was not an aim of the study, and the small sample size was likely insufficient to detect subtle differences between groups. The primary outcome measures were return to work rates at 6-month and 12-month follow-ups, and in a future trial, we would also use the text-based data collection system to collect information about actual return to work date. Determining the economic cost of the intervention is important given that although there is strong evidence of return to work interventions providing cost savings from a societal perspective, there is a need to ensure that actual intervention costs would be manageable once the intervention was implemented.[39] The outcomes for use in an economic analysis in a full trial were found to be feasible and acceptable. In particular, the tool for collecting data on work behaviour and GP attendance was successful, and the burden on participants did not appear to be excessive. In a future trial, we would also collect information about additional services that were used (eg, out of hours primary care services, minor injury units and counselling services), the number of hours/days of work that are missed (in this trial, we collected data on the number of days worked), as these could contribute to societal costs. Furthermore, the aim of the study was to explore the feasibility of undertaking an RCT of the WorkPlan intervention and this, along with the small sample size, meant that we did not examine the role of socioeconomic factors or type of employment as a mediator of returning to work in each of the groups. However, these data (including job title and postcode) were collected and could be used in a future trial.

Following on from the findings of the feasibility study, there are a number of adaptations proposed for a future study. A limitation of the current study is that the views of black and ethnic minority groups are under-represented. The recruitment materials and how patients are identified and approached will need refining to increase the number of men and black and ethnic minority participants. Attrition was higher among participants recruited from sites with research nurses rather than sites where the study research assistants recruited, which may be partly due to rapport and knowledge of the project, both

of which have been reported to impact recruitment.[40] To improve retention, particularly from sites where the participants are not recruited by the study research assistants, there needs to be greater effort to build rapport with participants and encourage retention in the trial. In addition, although we used incentives to encourage participation (including a £20 voucher for completing the assessment interview), it may be that greater consideration needs to be given to compensating participants for their time throughout the 12-month study. A further limitation is that the study did not include the views and experiences of employers, who may hold negative beliefs about the ability of cancer survivors to return to work[25] and therefore impact on the process of returning to work. Finally, although we examined engagement and acceptability of the intervention (through the interviews), we did not test the fidelity of the intervention as part of this feasibility trial. Therefore, in a full RCT, we would systematically assess intervention fidelity and, where possible, incorporate fidelity data in the analysis of outcomes.[41]

## CONCLUSION

This study investigated the feasibility of undertaking an RCT of a workbook-based intervention to support patients with cancer patients in returning to the workplace. The initial results are encouraging and suggest that the intervention was both well received and conveyed benefit to participants in supporting return to work after cancer treatment. The findings suggest that, with minor modifications, an effectiveness RCT is warranted.

**Author affiliations**
[1]Department of Psychological Sciences, Birkbeck University of London, London, UK
[2]Faculty of Health and Life Sciences, Coventry University, Coventry, UK
[3]Leeds Institute of Health Sciences, University of Leeds, Leeds, UK
[4]Oncology Department, Queen Elizabeth Hospital, University Hospitals Birmingham National Health Service Foundation Trust, Birmingham, UK
[5]Occupational and Environmental Medicine, Institute of Clinical Sciences, College of Medical and Dental Sciences, University of Birmingham, Birmingham, UK

**Contributors** EAG conceived the idea for the study. EAG, AJS and SSS contributed to the study design, randomisation and analysis plan. EAG wrote the first draft. LS, MA, PLW, PR, AJS, AZ and SSS were involved in multiple revisions. The final version of the manuscript was approved by all coauthors.

**Funding** This paper presents independent research funded by the National Institute for Health Research (NIHR) under its Research for Patient Benefit (RfPB) Programme (Grant Reference Number PB-PG-0613-31088).

**Disclaimer** The views expressed are those of the author(s) and not necessarily those of the NHS, the NIHR or the Department of Health and Social Care.

**Competing interests** None declared.

**Patient consent** Not required.

**Ethics approval** Ethical approval for this study was obtained from the National Research Ethics Service (Reference: 15/WM/0166) and research governance approval was obtained from all four participating NHS Trusts.

**Provenance and peer review** Not commissioned; externally peer reviewed.

**Data sharing statement** Additional unpublished data are not publicly available.

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
