## [Reviewer comments · BMJ Open]

ARTICLE DETAILS

TITLE (PROVISIONAL)	A feasibility randomized controlled trial of a guided workbook intervention to support work-related goals among cancer survivors in the UK
AUTHORS	Grunfeld, Elizabeth; Schumacher, Lauren; Armaou, Maria; Woods, Pernille; Rolf, Pauline; Sutton, Andrew; Zarkar, Anjali; Sadhra, Steven

VERSION 1 – REVIEW

REVIEWER	Gemma Morgan University of Bristol, UK
REVIEW RETURNED	24-Apr-2018

GENERAL COMMENTS	This paper describes a pilot RCT testing the feasibility and acceptability of the intervention and RCT study design. It is an interesting study and there is a clear argument for research in this area so a feasibility study on recruiting this population, and exploring acceptability of the new intervention is welcomed. However for me there are several major omissions from the manuscript: Recruitment / Retention A larger proportion of approached participants were deemed ineligible (n=170) than those deemed eligible (n=154) yet we are not given details of the reason for the ineligibility. Details of the recruitment processes are also rather limited, it would be of interest to the reader to understand how patients were approached in clinics - e.g. was it during the clinical appointment, at the start, at the end, was anyone else in the consultation room at the time the study was discussed, what sort of clinics were used for recruitment (e.g. are these treatment planning clinics, follow-up clinics, diagnostic clinics, etc?), were these clinics part of Foundation Trusts, community hospitals, tertiary care centres. The authors suggest that recruitment and adherence may have been affected by the type of recruiter but do not explain why and how a research nurse approach differs from a research study assistant approach in terms of building rapport and relationship. The data on attrition are presented as %ages and it is not clear what these pertain to and how attrition is defined (e.g. are these numbers not returned a text message response or numbers not completing a questionnaire or numbers withdrawing from the study?). A diagram describing the attrition at each time point and reason for attrition would help- understanding of attrition is an important feature of feasibility studies and one of the stated aims of the study so one would expect further detail on this to be provided. I find the term "pink collar workers" an unusual choice to describe a sociodemographic group and would suggest this term would be unfamiliar to many, at least UK-based, readers.
---

	The lack of heterogeneity amongst participants in terms of gender and ethnicity was well described. It would be helpful to know how non-representative this sample was compared to the population demographics in the catchment areas of the recruitment sites and whether better representation could be achieved by recruiting in areas of greater ethnic diversity. Outcomes It is stated that there were no adverse events during the pilot trial but no details are provided on how adverse event data were collected. I was interested and pleased to see data on healthcare use collected though it could be argued - perhaps something to explore in the definitive RCT - that wider health system demand would have been equally of interest (e.g. ED attendances, minor injury unit attendances, out-of-hours primary care services, and practice nurse appointments). Intervention It is alluded that this is a complex intervention yet the description of the intervention is very limited. Further detail on the content / length / format of the workbook should be provided. I appreciate the detail is published elsewhere but a more detailed summary should be included in this manuscript. The intervention theory is not mentioned. It was not clear why the intervention was prescribed at 4 weeks with a strict weekly regime - why were participants not permitted to engage with the workbook at a pace that suited them? How was this assessed? No comment was made on fidelity of the intervention and whether this was evaluated- i.e. what proportion of participants completed the workbook as the researchers intended? Results Despite the authors clearly stating - correctly - that hypothesis testing would be inappropriate given this is a feasibility study not a definitive RCT and no sample size calculation was conducted to ensure the study would be powered, p values are presented for t tests of differences in outcomes between intervention and control groups. Inclusion of such analyses is confusing and does not tell the reader anything about the effectiveness. There are a few grammatical errors in the text and some of the numbers in the CONSORT diagram are obscured.
--	---

REVIEWER	Alastair Munro St Andrews University United Kingdom
REVIEW RETURNED	03-Jun-2018

GENERAL COMMENTS	This is an important study which takes a straightforward approach to an increasingly common problem: the ability of patients with cancer to return to work after treatment is complete. Although the original protocol was published in the Journal of Internet Medical Research the authors have used a simple analogue approach, a workbook which is used to underpin a personalised return-to-work plan The paper is structured and presented well, and my criticisms and comments are minor in nature. Some of them concern points for consideration as the authors move on towards designing the full-scale randomised trial Page 5 line 19 I'm not sure what "building on communication and planning skills to brainstorm ways to achieve this goal" means
---

	Page 6 line 40 They did not stratify by gender. I appreciate that for breast, gynaecological, and prostate tumours, gender is not relevant. But it might be for genitourinary and bowel. page 6 line 49 it would have been helpful to have had an example of the work plan package placed with the supplementary material Page 7 line 14 it is a pity that it was not possible to gauge employers' views. It is quite possible that employers' reluctance to re-employ patients with cancer forms an important barrier. Page 7 line 34 if the actual date of return to work had been recorded it would have been possible to construct a Kaplan-Meier plot, with return to work as the event of interest. I suspect that this would have been more informative' and more powerful statistically' than simply enumerating reported rates at six months and 12 months post-treatment. Page 10 line 21 The higher attrition rate when research nurses had performed the initial recruitment is a provocative finding. Unfortunately, no numbers are provided so it is difficult to assess the magnitude of the effect. One possible implication is that human relationships are important here, and that the reason is that the research assistants were perceived as more invested in both the project itself, and in those who had agreed to participate. Page 10 line 51 The qualitative aspects of this study are dealt with in a somewhat cursory fashion. Is this because the authors plan to write a separate paper with details on the qualitative findings? Page 11 line 34 "the workbook format of the intervention was well received": where is the data to support this statement? Might not this finding simply be an example of acquiescence bias, particularly if the question asked was a leading one? Page 11 line 50 The problem of when treatment is completed is a perennial one. Patients may spend several years on adjuvant hormonal therapy. "Completed the intensive phase of treatment" might be better way of phrasing the question. Page 12 line 17 is "workability" the same as ability to work? Page 12 line 40/41 The figure quoted for return to work at 12 months in the intervention group (53%) is at variance with the abstract and the data in table 1 where the equivalent figure is given as 70% Page 12 One general weakness in this paper is the lack of more precise information on the socio-economic status of the participants and the precise nature of their employment. The local unemployment rate is a component of most indices of socio-economic deprivation and as such can be obtained via linkage to an individual's postcode. It is reasonable to ask whether, in areas of high unemployment, return to work is less likely. The colour classification for nature of employment that has been used (white, pink, blue) is ill-defined and I'm not sure how it works in practice. Would a healthcare assistant be pink collar and a state registered nurse be white-collar, or would they both be pink collar, or both be white collar?
--	--

	Page 14 line 56 It is interesting that the EQ-5D was perhaps not most suitable measure of health status. It is widely used in economic analyses of interventions in healthcare. Economic evaluation would be a key component of any full-scale studies arising from this feasibility assessment Page 15 there is no formal power calculation presented but on the basis of the results presented in this paper (baseline rate 47%; absolute rate difference 23%) the numbers required for a full randomised controlled trial to confirm this level of effect (alpha 0.05 power 0.8) are remarkably modest: 71 patients in each arm evaluable at 12 months. Table 1 comorbidities: how are these defined?
--	---

VERSION 1 – AUTHOR RESPONSE

Reviewer: 1

Gemma Morgan, University of Bristol, UK

Recruitment / Retention

A larger proportion of approached participants were deemed ineligible (n=170) than those deemed eligible (n=154) yet we are not given details of the reason for the ineligibility.

Reasons for ineligibility have now been included (page 10, paragraph 2).

Details of the recruitment processes are also rather limited, it would be of interest to the reader to understand how

patients were approached in clinics - e.g. was it during the clinical appointment, at the start, at the end, was anyone else in

the consultation room at the time the study was discussed, what sort of clinics were used for recruitment (e.g. are these

treatment planning clinics, follow-up clinics, diagnostic clinics, etc?), were these clinics part of Foundation Trusts,

community hospitals, tertiary care centres.

We have now included further detail of the recruitment setting and process (page 6; lines 19 to 24).

The authors suggest that recruitment and adherence may have been affected by the type of recruiter but do not explain

why and how a research nurse approach differs from a research study assistant approach in terms of building rapport and relationship.

We have now clarified this point (page 11; lines 10 to 13)

The data on attrition are presented as %ages and it is not clear what these pertain to and how attrition is defined (e.g. are

these numbers not returned a text message response or numbers not completing a questionnaire or numbers withdrawing

from the study?). A diagram describing the attrition at each time point and reason for attrition would help- understanding

of attrition is an important feature of feasibility studies and one of the stated aims of the study so one would expect further

detail on this to be provided.

We have now clarified that attrition was based on non-return of the questionnaire at each time-point (page 11, lines 5 to 8). We have now included reasons for attrition at each time-point in the study flowchart (Figure 1). I find the term "pink collar workers" an unusual choice to describe a sociodemographic group and would suggest this term would be unfamiliar to many, at least UK-based, readers. Pink collar refers more to occupational grouping than socio-demographic group – we have now removed this from the manuscript. The lack of heterogeneity amongst participants in terms of gender and ethnicity was well described. It would be helpful to know how non-representative this sample was compared to the population demographics in the catchment areas of the recruitment sites and whether better representation could be achieved by recruiting in areas of greater ethnic diversity. We have now added in further detail of the demographics of the geographic location where the majority of participants were recruited (page 14, paragraph 2, lines 5 to 9).

Outcomes

It is stated that there were no adverse events during the pilot trial but no details are provided on how adverse event data were collected. We have now included a paragraph about the process for collecting data on adverse events (page 8, final paragraph). I was interested and pleased to see data on healthcare use collected though it could be argued - perhaps something to explore in the definitive RCT - that wider health system demand would have been equally of interest (e.g. ED attendances, minor injury unit attendances, out-of-hours primary care services, and practice nurse appointments). This is an interesting point and something would be valuable to include in a future trial - we have mentioned this now on page 15 (paragraph two, lines 12 to 14).

Intervention

It is alluded that this is a complex intervention yet the description of the intervention is very limited. Further detail on the content / length / format of the workbook should be provided. I appreciate the detail is published elsewhere but a more detailed summary should be included in this manuscript. Previously we referred readers to published protocol paper for this trial but we have now included additional detail about the intervention in this manuscript also on page 7 (paragraph 1). The intervention theory is not mentioned. We have now added in further detail about the theories that were outlined in the intervention development (page 7, paragraph 1). It was not clear why the intervention was prescribed at 4 weeks with a strict weekly regime - why were participants not permitted to engage with the workbook at a pace that suited them? How was this assessed? We have now clarified that the four-week period was suggested rather than imposed and participants could/did work through the workbook at their own pace (page 7, paragraph 2). No comment was made on fidelity of the intervention and whether this was evaluated- i.e. what proportion of participants completed the workbook as the researchers intended?

We did not examine the fidelity of the intervention as part of this feasibility trial but accepted that this would be valuable to undertake as part of a full trial. We have added this under limitations in the discussion section (page 16, paragraph 1).

Results

Despite the authors clearly stating - correctly - that hypothesis testing would be inappropriate given this is a feasibility study not a definitive RCT and no sample size calculation was conducted to ensure the study would be powered, p values are presented for t tests of differences in outcomes between intervention and control groups. Inclusion of such analyses is confusing and does not tell the reader anything about the effectiveness.

We have now removed these analyses from the manuscript.

There are a few grammatical errors in the text and some of the numbers in the CONSORT diagram are obscured.

Thank you for pointing this out – these points have now been corrected.

Reviewer: 2

Alastair Munro, St Andrews University, United Kingdom

Page 5 line 19 I'm not sure what "building on communication and planning skills to brainstorm ways to achieve this goal"

means

We have now rephrased this sentence to be clearer for the reader (page 5, paragraph 2)

Page 6 line 40 They did not stratify by gender. I appreciate that for breast, gynaecological, and prostate tumours, gender is not relevant. But it might be for genitourinary and bowel.

It was not felt to be feasible given that only one of the four cancer types were mixed gender. However, we have now mentioned

stratifying by age in a larger trial incorporating a greater range of cancer types.

page 6 line 49 it would have been helpful to have had an example of the work plan package placed with the supplementary

material

We are still seeking funding for the full trial and so plan to do this when we publish the full trial. We have added further detail of

the intervention on page 7 (paragraph 1).

Page 7 line 14 it is a pity that it was not possible to gauge employers' views. It is quite possible that employers' reluctance

to re-employ patients with cancer forms an important barrier.

Yes, although our pilot work indicated that participants did not want employers involved until after they have completed the

intervention. We have acknowledged that this is a limitation on page 15 (final paragraph)

Page 7 line 34 if the actual date of return to work had been recorded it would have been possible to construct a KaplanMeier

plot, with return to work as the event of interest. I suspect that this would have been more informative' and more

powerful statistically' than simply enumerating reported rates at six months and 12 months post-treatment.

A Kaplan-Meier plot was outside the scope of this feasibility study but would be valuable to include in a future study.

Page 10 line 21 The higher attrition rate when research nurses had performed the initial recruitment is a provocative

finding. Unfortunately, no numbers are provided so it is difficult to assess the magnitude of the effect.

One possible

implication is that human relationships are important here, and that the reason is that the research assistants were perceived as more invested in both the project itself, and in those who had agreed to participate. We have added in further detail of possible reasons for this finding on pages 11 (second paragraph) and page 15 (final paragraph).

Page 10 line 51 The qualitative aspects of this study are dealt with in a somewhat cursory fashion. Is this because the authors plan to write a separate paper with details on the qualitative findings?

Yes, that is correct. There is not the space in this manuscript to report the qualitative findings in full and we have reported these in the following paper which we refer the reader to (page 9, qualitative data paragraph).

Schumacher L, Armaou M, Rolf P, Sadhra S, Sutton AJ, Zarkar A, Grunfeld EA. Usefulness and engagement with a guided workbook intervention (WorkPlan) to support work related goals among cancer survivors. *BMC Psychol.* 2017;5(1):34

Page 11 line 34 "the workbook format of the intervention was well received": where is the data to support this statement?

Might not this finding simply be an example of acquiescence bias, particularly if the question asked was a leading one?

We have now included acquiescence bias as a possible interpretation in our discussion section (page 15, first paragraph on the discussion).

Page 11 line 50 The problem of when treatment is completed is a perennial one. Patients may spend several years on adjuvant hormonal therapy. "Completed the intensive phase of treatment" might be better way of phrasing the question.

We have rephrased this as suggested by the reviewer.

Page 12 line 17 is "workability" the same as ability to work?

Workability is the individual's perception of their ability to work and we have now clarified this on page 4 (final paragraph)

Page 12 line 40/41 The figure quoted for return to work at 12 months in the intervention group (53%) is at variance with

the abstract and the data in table 1 where the equivalent figure is given as 70%

Thank you for spotting this and we have now corrected through the manuscript.

Page 12 One general weakness in this paper is the lack of more precise information on the socio-economic status of the

participants and the precise nature of their employment. The local unemployment rate is a component of most indices of

socio-economic deprivation and as such can be obtained via linkage to an individual's postcode. It is reasonable to ask

whether, in areas of high unemployment, return to work is less likely.

The majority of this data collection took place in Birmingham. In a full trial we plan to recruit nationwide and would therefore the

reviewers comment regarding local unemployment rates and socioeconomic status will be of value in that study (page 15, paragraph 2).

The colour classification for nature of employment that has been used (white, pink, blue) is ill-defined and I'm not sure

how it works in practice. Would a healthcare assistant be pink collar and a state registered nurse be white-collar, or would

they both be pink collar, or both be white collar?

The first reviewer also did not like the use of the pink, blue and white collar classification system and so we have removed from this manuscript

Page 14 line 56 It is interesting that the EQ-5D was perhaps not most suitable measure of health status. It is widely used in economic analyses of interventions in healthcare. Economic evaluation would be a key component of any full-scale studies arising from this feasibility assessment

We have removed this statement as both the study health economist and a clinical trials unit we have worked with have agreed that the EQ-5D would need to be used in a full trial

Page 15 there is no formal power calculation presented but on the basis of the results presented in this paper (baseline rate 47%; absolute rate difference 23%) the numbers required for a full randomised controlled trial to confirm this level of effect (alpha 0.05 power 0.8) are remarkably modest: 71 patients in each arm evaluable at 12 months. We did not aim to use the feasibility results as the basis for a power calculation for a full trial and so have not reported that in this paper. We have worked with a clinical trials unit to develop the plan for the full RCT and as we are opening up the range of cancer types (who have varying return to work rates) they have used an estimate of 15% not returning to work within 12 months.

As such the sample size calculation is as follows: assuming 90% power and 5% significance, 15% not returning to work within a year, 25% drop out and a hazard ratio of 1.5 gives a total sample size of 402 (201 per group).

Table 1 comorbidities: how are these defined?

We have now included a definition of co-morbidities as applied in this study (page 10, final paragraph)

VERSION 2 – REVIEW

REVIEWER	Alastair Munro University of St Andrews United Kingdom
REVIEW RETURNED	12-Jul-2018
GENERAL COMMENTS	The authors have addressed all my concerns - this is an excellent paper describing an original and interesting piece of work.